# Nursing Interventions against Bullying: A Systematic Review

**DOI:** 10.3390/ijerph20042914

**Published:** 2023-02-07

**Authors:** María del Carmen Celdrán-Navarro, César Leal-Costa, María Suárez-Cortés, Alonso Molina-Rodríguez, Ismael Jiménez-Ruiz

**Affiliations:** 1Faculty of Nursing, University of Murcia, El Palmar, 30120 Murcia, Spain; 2ENFERAVANZA, Murcia Institute for BioHealth Research (IMIB-Arrixaca), El Palmar, 30120 Murcia, Spain

**Keywords:** nursing, intervention, bullying, intimidation

## Abstract

(1) Background: Bullying is a worldwide public health problem, with short- and long-term physical, mental, and socio-economic implications for all involved, including consequences as serious as suicide. (2) Objective: The aim of this study is to compile data on nursing interventions for preventing and addressing bullying at the international level. (3) Methods: A systematic review was conducted in accordance with the guidelines laid out in the PRISMA statement. The search included papers written in Spanish, English, and Portuguese over the previous five years from the following databases: Web of Science, CUIDEN, CINHAL, BDENF, Cochrane, Lilacs, and PubMed. The following descriptors were used: “Acoso escolar AND Enfermería”, “Bullying AND Nursing” and “Intimidação AND Enferma-gem”. Due to the heterogeneity in the methodology of the studies, a narrative synthesis of the results is provided. (4) The synthesis of results shows nurses’ involvement in tackling and preventing bullying. Interventions are categorised into awareness raising; coping mechanisms; and approach/care, nursing skills in the face of bullying, and the role of the family in the face of bullying. (5) Conclusions: It is clear that at the international level, nursing is involved in planning and developing autonomous and interdisciplinary interventions to address and prevent bullying. The evidence paves the way for school nurses and family and community nurses to take steps to tackle this phenomenon.

## 1. Introduction

Nowadays, school bullying, henceforth known as bullying, is a worldwide public health problem [1,2,3]. It is defined as intentional and repeated violence committed by the perpetrator(s) towards a peer in an educational context, exploiting a physical, psychological, or social power imbalance, with the peer becoming a habitual victim [4,5,6,7,8]. This phenomenon is also hard to detect thanks to the so-called triple law of silence, which normalises it and renders it invisible [4].

To make matters worse, with the advent of relationship, information, and communication technologies and the use of social networks (RRSS), there has been a shift in the way we, and especially young people, relate to each other. As a result, bullying is now taking place in a wider context, giving rise to cyberbullying. Bolstered by anonymity and widespread Internet use on personal computers and smartphones, cyberbullying further exacerbates this situation [5,9,10]. 

Regardless of the medium used to express this phenomenon, violence can have repercussions for the health of all those involved, be they victims, perpetrators, or bystanders, which can become apparent at the time of the aggression and/or in the long term [4,5]. These may include short-term mental health problems such as obsessive disorders, anxiety-depressive disorders, sleep disorders, phobias, post-traumatic stress “flashbacks”, addictions, low self-esteem, aggression, and emotional exhaustion. Physical and psychosomatic problems such as somatisation and injuries of varying severity and type also feature prominently. Another important factor is the socio-economic impact, which can cause deficiencies in social skills and relationships (isolation), as well as academic failure due to underachievement and absenteeism. In the long term, such mental health problems can become chronic, leading to more complex issues. The victim is at risk of perpetuating their role in future relationships (e.g., gender-based violence, mobbing, etc.) and of earning a lower income because of poor employment status, linked to dropping out of school. In both the short and long term, this can have serious repercussions, including suicide [1,4,6,11]. 

In the case of bullies, there is evidence that they may perform less well academically and experience relational problems as a result of reduced empathy, increased cruelty, insensitivity, and a disregard for rules. Over time, they could run the risk of lower earnings through job failure, legal problems due to criminal behaviour and substance abuse, in addition to continuing to play the role of perpetrator, through a pattern of dominance–submission (gender-based violence) [3,5,7]. 

The scale of this phenomenon in Spain stands at a victimisation rate of around 15.4% [10]. This is in line with a study conducted in the north of the country, where the prevalence regardless of sex is 12% for traditional bullying, 8.1% for victims of cyberbullying, and 7% for cyberbullies. As a further point relating to the scale of the problem, one of these publications indicates that in 3% and 6.6% of cases, respectively, cyberbullying and traditional bullying lead to serious health problems [12]. This research reveals that 46.4% of students engage in name-calling or ridicule, 46.8% use insults and threats, and 31.1% admit to not intervening when they witness bullying. A total of 68% of teachers say that insults are very frequent, 54% report physical aggression, 38% report theft, 74% hear name-calling on a daily basis, and 72% hear false rumours. Almost half of teachers (48%) feel unable to deal with problems arising from bullying. This is in stark contrast to the views of parents, all of whom believe that their children are not bullies, while 80% are of the opinion that bullying is not common in schools [10,12]. 

Given this state of affairs, many professionals are involved in the search for a solution to this problem. Among them are nurses who, by virtue of their expertise and ethics, have a social responsibility to protect the rights of children by ensuring their dignified and healthy growth. As such, nurses have the most important role to play in this area [13]. 

Similarly, in their research on the partnership between school psychologists and school nurses, Kub and Feldman argue that “*it is the responsibility of both groups to work collaboratively to address bullying as a school health issue*” (2019) [14]. In addition, and keeping with the theme of identifying the competencies of school nurses, a recent Spanish paper makes the training and development of school nurses in the field of bullying a priority [15]. 

In view of the prevalence data and implications for health, and to draw attention to nurses’ involvement in this phenomenon, we have posed the following research question. What are the preventive measures and approaches that nurses are taking against bullying at the international level? This is addressed through the objective of this research to compile data on nursing interventions for preventing and addressing bullying at the international level. 

## 2. Materials and Methods

A systematic review was conducted according to the guidelines set out in the PRISMA statement [16]. As this includes research of different types (both quantitative and qualitative), a narrative review was chosen as the most appropriate way to synthesise the results in this case [17]. The protocol was first registered in PROSPERO (International Prospective Register of Ongoing Systematic Reviews) under registration number ID:CRD42023393229.

### 2.1. Search Strategy

The literature search was carried out in the following databases: Web of Science (WOS), CUIDEN, CINHAL, BDENF, Cochrane, Lilacs, and PubMed. The search was carried out in October 2022, with the database last accessed on 4 October 2022, in English, Portuguese, and Spanish. The following descriptors were used in combination with Boolean operators: “Acoso escolar AND Enfermería”, “Bullying AND Nursing” and “Intimidação AND Enfermagem”. For CINHAL, WOS, and PubMed, the exclusion Boolean “NOT mobbing” was added, as the search yielded a large number of studies on workplace bullying within nursing. This led to a very high number of potentially ineligible papers, which in turn made it difficult to review the records retrieved. 

### 2.2. Inclusion and Exclusion Criteria

Studies on nursing interventions to address bullying published in Spanish, English, and Portuguese were included. The time frame was limited to the previous five years. In terms of the type of studies, both qualitative and quantitative research was included, reporting on one or more interventions against bullying and involving one or more nurses in different care settings (outpatient, school, and inpatient). No restrictions were placed on the number of professionals involved in the intervention or on the age of the recipients. 

Studies in which the interventions did not involve a nursing professional (in any role or level of responsibility) were excluded, as were systemic reviews of nursing interventions. 

### 2.3. Study Selection

Duplicates were removed after each search with the help of the Mendeley Desktop application. The first phase involved screening by reading the abstract/title. In the second phase, two independent reviewers read the full text of the pre-selected studies, coming to a consensus on a new list of potentially eligible articles. There were no discrepancies between reviewers. The titles of the bibliographic references were also examined, with the addition of two papers that met the aforementioned criteria but had been omitted during the search process.

### 2.4. Risk of Bias Analysis

After full text reading, a total of 17 selected articles were analysed for bias. As the type of article varied widely, we used the following checklists depending on the study design: CONSORT (controlled experimental and quasi-experimental studies), COREQ (qualitative studies), and STROBE (observational studies). The cut-off threshold was set as the mean value of each scale (50% of the items). This had to be exceeded as a criterion for eligibility. Although there were only minor discrepancies between the two reviewers, concordance between scores was calculated using Cohen’s kappa coefficient (*p* < 0.000). This means that the measures were concordant, with variation occurring due to chance. 

### 2.5. Tabulation and Data Analysis

The lead author compiled the studies eventually included in the systematic review in an Excel database. This was reviewed by a second author, with any uncertainties being resolved between the two. Lastly, the results tables were prepared to summarise the data from each of the selected studies in accordance with the methodology.

### 2.6. Data Synthesis

Due to the disparity in the outcome measures of the quantitative studies, we decided not to conduct a meta-analysis. On this occasion, a qualitative synthesis of the results was performed, summarising the characteristics of the articles (study population, intervention, measures, and conclusions) using descriptive statistics.

## 3. Results

As shown in Figure 1, the search initially returned 1371 publications, 84 of which were rejected as duplicates. After reading the title and abstract, 1153 titles were excluded, leaving 59. In the second phase, 50 articles were discarded for various reasons after reading the full text: access to the full text was not available (2 articles) and the intervention was not led or delivered by nurses (48 studies). The risk of bias analysis did not lead to the removal of any studies, with all exceeding the threshold. In the end, 17 studies remained for inclusion in this systematic review, 8 with a qualitative design and 9 using quantitative methodology. 

### 3.1. Study Characteristics

Eight qualitative articles were included. These were published between 2018 and 2022. Study participants ranged in age from 11 to 17 years old. The mean was not calculated as not all studies reported this data. Sample sizes ranged from 12 to 134 students and from 11 to 12 school nurses. The study designs and methodologies included were as follows: exploratory, phenomenological, and hermeneutic with interviews and focus groups (n = 5), exploratory using Paulo Freire’s culture circles (n = 2), and rapid qualitative research (n = 1).

There were also nine quantitative articles. These were published between 2019 and 2022, with study participants ranging in age from 10 to 16 years old. The mean was not calculated, since some studies did not report it, and also due to the heterogeneity of the sample components. Sample sizes for the quantitative articles ranged from 68 to 21,075 children, with 40 school nurses, 40 school administrators, and 10 teachers involved. The designs and methodologies of the studies included were experimental randomised clinical trial (n = 1), quasi-experimental pre-post intervention (n = 3), case-control (n = 2), and observational (n = 2).

### 3.2. Summary of Results

After tabulating (Table 1 and Table 2) the interventions found in the articles, five categories were created covering the basic features of and types of interventions related to bullying: Primary Prevention: Awareness raising; Secondary Prevention: Coping mechanisms; Tertiary Prevention: Approach/Care, nursing skills in dealing with bullying, and role of the family in dealing with bullying. In addition, levels of evidence for the interventions were included according to the Scottish Intercollegiate Guidelines Network (SIGN) methodology [18]. SIGN establishes a classification of the level of evidence of the studies reporting the level of evidence, the degree or strength of recommendation for the development of clinical practice guidelines or interventions. SIGN has eight levels of evidence (1++, 1+, 1−, 2++, 2+, 2−, 3, and 4). Studies classified as 1− and 2− should not be used in the process of developing recommendations because of the high possibility of bias.

## 4. Discussion

### 4.1. Primary Prevention: Awareness Raising

Three of the articles [4,6,25] report on strategies for primary prevention through awareness raising. These are based on the provision of support through social influence and peer mentoring, introduced through fun activities such as murals, songs with positive messages [4], dramatization [4,6], and watching short films [25]. All seek to create a participative and reflective group atmosphere that helps to prevent bullying. Brandão et al. [4] stated that it is, “*the school students themselves who can take on this role of caring for others*”, noting that this “*opened space to potentialize resilience, broaden active listening and give a voice so adolescents can act as managers of their processes of self-discoveries”* (2020). These data are consistent with research by Byrne et al. [26] in which peer mentoring is shown to play a preventive role in adolescent inter-relational conflict situations (2018). 

### 4.2. Secondary Prevention: Coping Mechanisms

Knowing how to prevent bullying from happening is just as important as being prepared to deal with it. Six of the articles [1,2,7,9,19,22] deal with interventions aimed at tackling bullying. To this end, nurses enter the realm of those affected by this phenomenon, through social networks and web radio programmes [9], in order to gain an understanding of its significance, perspectives on the problem, and possible options for addressing it [1,7,9,19]. Similarly, in three of the articles, nurses base their interventions on empathy and emotional recognition, being accompanied, talking to an adult, and shifting the focus of attention [2,7,22]. One of the pieces of research is a comprehensive three-phase nursing care process involving victims and bullies and using the solution-focused approach. This aims to increase the knowledge of those involved about behaviours, the effects of bullying, coping skills, and individual and collective problem solving, as well as sharing the experiences of bullying [7]. Similar research found that emotional coaching of children affected by school violence using the solution-focused approach methodology could be effective in reducing perpetration and victimisation [27,28].

### 4.3. Tertiary Prevention: Approach/Care

This category includes two publications [3,11] detailing nursing care interventions intended to mitigate the aforementioned negative repercussions that bullying can have on the physical, psychological, or socio-economic health of those involved. The focus is on group techniques, the solution-focused approach [11], and cognitive-behavioural therapy, supported by fun activities such as dramatization, games, and role-play [3]. In both pieces of research, support groups aim to improve the social skills, empathy, and resilience of bullying victims [3,11]. As mentioned earlier in this article, the solution-focused approach could be useful as a secondary prevention, as well as a basis for providing nursing care. Improvements could be observed in bullying victims’ social sphere, in particular with noticeable effects on their well-being and coexistence in the school environment [27,28]. 

### 4.4. Nursing Skills in Dealing with Bullying

Four of the studies deal with practice, skills, and knowledge from nurses’ perspectives on bullying [5,8,20,23]. Three strategies emerge from the actions described in these articles: 1. increasing the hours of direct school nurse involvement (decreasing ratios) in students’ psychosocial sphere, logging and processing information from cases of bullying, and continuous training on bullying [23]; 2. collating school nurses’ expertise on bullying concepts, such as how to handle cases, how to apply protocols and guidelines, and prevention [5,8]; and 3. comparing school nurses’ perceptions of LGBTQ bullying with those of victims [20]. It is essential to gain an understanding of the situation of the healthcare personnel who care for this group, in order to obtain data that will facilitate training to improve the quality of care and attention [8,29]. As shown by recent research findings that also measure practices, skills, and knowledge on bullying in other groups, opportunities to detect and tackle bullying are being missed due to gaps in the training of the professionals involved [29]. 

### 4.5. Role of the Family in Dealing with Bullying

In two of the studies, nurses examine the family’s potential involvement in bullying [21,24]. These studies seek to involve the children’s immediate social circle in an effort to respond at all levels of prevention, awareness, and sensitisation; the recognition of initial psychological and psychosomatic symptoms (e.g., sadness, anger); the provision of advice and family motivational support; family strategies to speed up recovery; and spiritual support by means of cultural practices and respect for beliefs (e.g., prayers, chanting, faith-building). Also included is the enhancement of social relations in the place of residence (neighbourhood) to provide a source of support that lessens the impact of bullying. Lastly, it is also important to work on the resilience of bullying victims’ entire family unit [21,24]. These findings are consistent with the results obtained by Shaheen et al. [30], “*Given the importance of socio-familial support, there is a need to increase nursing interventions that focus on family-centred care to protect those affected by bullying dynamics*” (2019).

### 4.6. Limitations

The high level of heterogeneity found among the outcome measures of the quantitative studies meant that it was not possible to perform a meta-analysis in accordance with the recommendations of the PRISMA statement. 

The levels of evidence for interventions are generally low due to the types of study design used, making it difficult to make the case for their inclusion in programmes aimed at tackling this issue. One task that remains pending for nurses is to measure the effectiveness of their actions. 

## 5. Conclusions

The results show that nurses carry out anti-bullying interventions at various levels of prevention, ranging from awareness-raising, sensitisation, detection, and referral to directly tackling the problem in its most acute phase through nursing care, using methods such as behavioural therapy, emotional expression, counselling, support, relational skills training, and empowerment. 

These actions are primarily aimed at students, encompassing all roles in the bullying dynamic (victim, perpetrator, or bystander). They also involve the school community and parents, tackling the situation from an individual and/or group perspective, creating a shared outlook, fostering a climate of safety and participation, and making those involved feel that the nurse can be a reliable and trustworthy resource, with the right tools and operating in accordance with professional ethical and deontological standards. In addition to working alone, the nurse may be able to judge when there is a need for referral to other professionals for more specialised treatment and even work holistically with other professionals to provide integrated care for all those affected by bullying. 

Nursing is therefore uniquely placed from a primary care perspective to provide the knowledge and skills necessary for the development of anti-bullying interventions, as demonstrated by the studies of Yosep [31,32]. However, further research providing solid scientific evidence and outcome measures that can be compared in future meta-analytical reviews is required.

## Figures and Tables

**Figure 1 ijerph-20-02914-f001:**
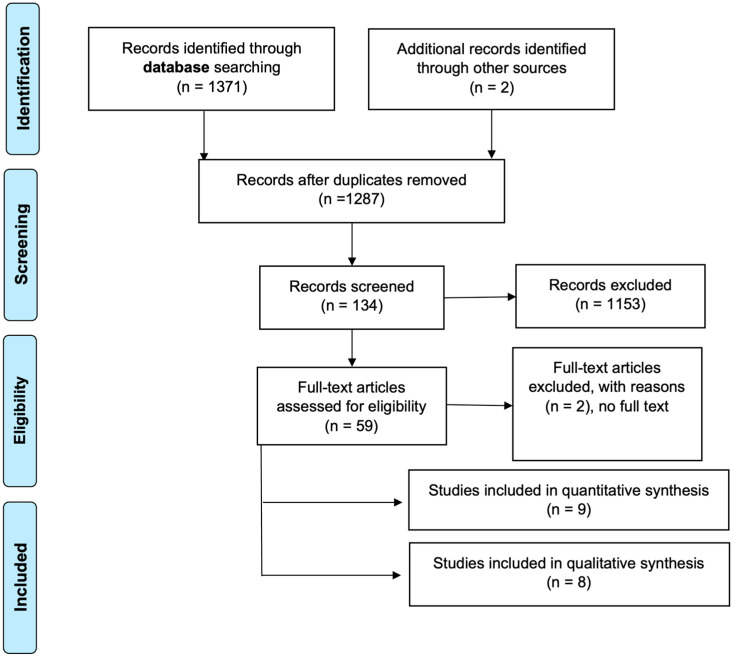
PRISMA flowchart.

**Table 1 ijerph-20-02914-t001:** Synthesis of interventions from qualitative studies and Scottish Intercollegiate Guidelines Network (SIGN) levels of evidence.

Authorship/Year/Country	Target Group/Performed by/Duration	Procedure/Outcome/Type of Intervention	SIGN
Brandão-Neto W et al. (2020), Brazil [4]	Students aged 13–16 Researchers (nurses) in collaboration with teaching staff 1 year (June 2017–May 2018)	Anti-bullying Health Education Programme (PATES). Selection/training of influential pupils with high social status, leadership, and conflict resolution skills. Culture circles, with themes: Role of a protagonist adolescent faced with bullying. Method: theatre. Reflection: importance of the protagonist adolescent in preventing and making positive changes in the face of bullying. Prevention and coping. Method: mural. Reflection: main actions and potential of the school context. Actions carried out: theatre, rap, debate videos, competitions, posters, and spaces of free expression. - Primary Prevention: Awareness raising	3
Abreu LDP et al. (2020), Brazil [9]	Young people aged 14–15 Researchers (nurses), teaching staff, a nurse specialist, and radio personnel 1 h	Web radio “In Tune with Health” programme, interview with a nurse specialising in cyberbullying. Discussion based on the anchor question “What do you understand by cyberbullying?”, via social media, the radio programme’s web board, Whatsapp, and Facebook, leading to: Definition of cyberbullying. Consequences of cyberbullying: health of all those involved. Coping strategies within the school environment. Knowledge of anti-bullying legislation. - Secondary Prevention: Coping mechanisms	3
Alvarado Romero HR et al. (2021), Colombia [19]	Adolescents aged 12–17 Researchers (nurses) Not specified	Analysis of participants’ knowledge of bullying via a survey and focus group discussion. Results: Definition of bullying. Causes: why bullying takes place. Reaction: coping with bullying. Where: places in the school environment where bullying takes place. - Secondary Prevention: Coping mechanisms	3
Heitmann AJ, Valla L, Kvarme LG, (2022), Norway [11]	Young people aged 11–13 Researchers (nurses), school nurses and teachers 4 months (January–April 2021)	Support groups (60 min) for victims of bullying based on the solution-focused approach. Three teachers and one school nurse received four days of training in group management and solution-focused approach techniques. The groups and interviews were structured using ad hoc guides. Results: Following the intervention, 40-min interviews were conducted with four participants who had experienced bullying. In addition, weekly 30 min follow-up sessions were held in which they expressed how effective their participation in the support groups had been in coping with their bullying situation. There was some improvement among bullying victims. - Tertiary Prevention: Approach/Care	3
Kvarme LG, Misvaer N, Valla L, Myhre MC, Holen S. (2019) Norway [1]	Students aged 13–14 Researchers (nurses) and 4 school nurses 1 month (December 2017)	Pilot project “School Health” which included a web-based questionnaire completed before a consultation with the school nurse, who, after analysing the information from the questionnaires, decided who would take part in the individual interviews and focus groups. A guide was available for both processes. Results: Talking about bullying is difficult. The school nurse is an important ally against bullying. The importance of social resources (friends). - Secondary Prevention: Coping mechanisms	3
Cohen SS, Grunin L, Guetterman TC. (2022) USA [5]	School nurses Researchers (nurses) 10 months (January–October 2019)	45 min telephone interviews with school nurses about their role in dealing with bullying. Results: Knowledge and implementation of the DASA (Dignity for All Students Act) in the USA. Bullying related to chronic health conditions. Behavioural health problems and bullying. Implementation of school-based programmes and policies for preventing and addressing the problem. New opportunities for nurses to engage with bullying victims with chronic health conditions. - Nursing skills in dealing with bullying	3
Earnshaw VA et al. (2021) USA [20]	Students aged 13–24 and school nurses Researchers (nurses) Not specified	Five online asynchronous focus groups were held to find out how nurses engage with and perceive the detection and management of bullying of LGBTQ students, compared to the experience of LGBTQ students. Rapid qualitative inquiry methodology was followed. Results: There is a disconnect between students’ perceptions of LGBTQ bullying and those of school nurses. - Nursing skills in dealing with bullying	
Soimah AYS, Hamidÿ NCD (2019) Indonesia [21]	Parents of bullying victims, aged 38–56 Researchers (nurses) 6 months (January–June 2017)	Personal interviews at the participants’ homes, with data obtained using the Colaizzi method. Results: Early responses to bullying (sadness and anger). Providing advice and motivation as a form of family support for bullying victims. Family support strategies to speed up the recovery process: spiritual support (prayers, chanting, etc.). Social resources as a source of support to help overcome the impact of bullying. Family resilience among bullying victims. - Role of the family in dealing with bullying	3

**Table 2 ijerph-20-02914-t002:** Synthesis of interventions from quantitative and observational studies and Scottish Intercollegiate Guidelines Network (SIGN) levels of evidence.

Authorship/Year/Country	Target Group/Performed by/Duration	Procedure/Outcome/Type of Intervention	SIGN
Avşar F, Alkaya S, (2019), Turkey [22]	Children aged 10–11 Researchers (nurses) Not specified	Validation of the Student-Advocates scale, which is based on four useful strategies to support bystanders in bullying situations. Results: Stealing the show, shifting the focus of attention. Turning it over, asking an adult for help. Being accompanied, victims and bystanders. Coaching in compassion for perpetrators. - Secondary Prevention: Coping mechanisms	2++
Alencastro LCS et al. (2020), Brazil [6]	Pupils aged 15–16 Researchers (nurses) 2 months (October–November 2016)	Anti-bullying intervention based on the “Theatre of the Oppressed”, divided into 16 sessions, 4 team meetings, 10 rehearsals, and 2 theatre performances. Results: Comparison of the difference in victimisation in the intervention group (IG) and control group (CG). In the IG, there was a reduction in the incidence of any of the types of aggression, compared to the CG. - Primary Prevention: Awareness raising	2++
Öztürk Çopur E, Kubilay G, (2021) Turkey [7]	Adolescents aged 13–14 Researchers (nurses) 6 weeks	This applies the solution-focused approach as a method of social skills training for dealing with bullying, with its impact being measured using the Personal Information Form and the Adolescent Peer Relationship Instrument (APRI) scale, with weekly 50 min sessions around themes: Recognising bullying behaviours. Effects of bullying and coping skills. Setting goals in the face of bullying. Understanding victims’ emotions and empathising and reflecting on solutions. Sharing bullying experiences and revealing exceptions. Recognising working mechanisms, highlighting strengths, and remembering individual goals. Results: Decrease in victimisation and aggression in the intervention group. - Secondary Prevention: Coping mechanisms	2+
Federici RA et al. (2020) Norway [23]	School nurses, school principals, and the municipal authority Researchers (teachers and psychiatrists) 5 years: 3 interventions, 2 follow-ups	Increase the presence of school nurses by 50%, to six and a half hours (three and a quarter hours for student care and three and a quarter hours for administrative and training tasks) in 5th, 6th, and 7th grades, reducing ratios (maximum two students per nurse) and attending to the students’ psychosocial environment, in order to reduce bullying. Beforehand, the researchers organised a two-day briefing session for the school health service, school nurses, school headteachers, and the municipal authority. Expected results: improvement in the psychosocial environment and academic results. - Nursing skills in dealing with bullying	1+
Karataş H, Öztürk C (2019) Turkey [24]	113 pupils and 26 parents Researchers (nurses) 1 year 7 weeks	Three-tiered anti-bullying training: - Students: Five weekly 40 min sessions, with awareness-raising and conflict resolution techniques, using slides, videos, pictures, scenarios, mnemonic games, acting roles, questions-answer activities, and the preparation and presentation of posters. - Parents: Two 60-min seminars: awareness and symptomology. - Teachers: One 60-min seminar with topics: anti-bullying awareness, negative consequences, proposed activities for prevention in the school environment, and possible improvements in teachers’ attitudes. Data were collected at four points in time: before the training, and two weeks, six months, and one year after the training, using the “Peer Bully Scale-Adolescent” questionnaire. Results: effective in the medium/long term for victimisation, but not for perpetration. - Secondary Prevention: Coping mechanisms	2+
Evgin, D., Bayat, M, (2020) Turkey [2]	Adolescents aged 12–14 Researchers (nurses) 5 months (May–September 2014)	Three-stage nursing care process, adopting Johnson’s Behavioural System educational model: Prevalence of bullying in the control group (CG) and intervention group (IG). Nine creative drama sessions, methodology: role play, improvisation, brainstorming, information cards, photographic memory. Sessions (S1–S9): S1: Introduction and rules. S2: Awareness and emotion recognition. S3: Concepts of bullying. S4: Empathy towards victims. S5: Problem solving skills. S6, 7, 8: Resolution of bullying situations. S9: Suggestions for dealing with bullying. Effectiveness was measured at two points in time, one week and three months afterwards using the tools: Traditional Peer Bullying Scale (TPBS), Problem-Solving Inventory for Children (PSIC), and Empathy Index for Children (EIC). Results: Providing health education via this model is effective for alleviating bullying in the short and medium term. Moreover, it helped to define the nurse’s role in bullying: protection (definition of risk factors), support (stimulation, information about bullying and what to do when you are a victim), and referral. - Secondary Prevention: Coping mechanisms	2+
Silva JL et al. (2018), Brazil [3]	Students with an average age of 11.2 years 4 months (March–June 2015)	Social skills training over eight weekly 50 min sessions, based on cognitive-behavioural techniques: role-playing, dramatizations, positive reinforcement, modelling, feedback, videos, and homework. Sessions were structured into three sections: 1: explanation of the task, 2: performance of the task, 3: feedback on the task. Included content and activities on good manners, making friends, empathy, self-control, emotional expressiveness, assertiveness, and interpersonal problem solving. Results: a reduction in victimisation was observed in the intervention group. - Tertiary Prevention: Approach/Care	2−
Hutson E, Melnyk B, Hensley V, Sinnott LT, (2019) USA [8]	Paediatric primary care nurses Researchers (nurses) 1 month (May 2017)	Description of paediatric primary care nurses’ practices, knowledge, and attitudes using the modified Health Care Provider’s Practices, Attitudes, Self-Confidence and Knowledge Regarding Bullying (HCP-PACK) questionnaire. Results: A high percentage (69%) does intervene in bullying situations. Almost all (90%) support taking action on a daily basis from primary care, including registering cases; making referrals to mental health, social work or psychology; counselling; providing families with anti-bullying materials and resources; reporting to the school board or directing parents to do so; consulting laws on bullying; advocating for change of school; and treatment of injuries. - Nursing skills in dealing with bullying	2++
Burk J, Park M, Saewyc EM. (2018), Canada [25]	Adolescents aged 13–18 Researchers (nurses) 6 months (January–June 2013)	“Out in Schools” programme of short films featuring different sexual orientations, aiming to reduce homophobia, biphobia, and transphobia, leading to a lower rate of bullying and suicidal tendencies and an increase in support from the school environment. Sessions of one–two hours. Data were acquired from the British Columbia Adolescent Health Survey (BCAHS). Results: programme linked to an improvement in the well-being of LGBTQ students. - Primary Prevention: Awareness raising	3

## Data Availability

The data are available upon email request to the corresponding authors.

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
