# Peer review of "Nursing Interventions against Bullying: A Systematic Review"

_ijerph, 2023, doi:10.3390/ijerph20042914_

Round 1

Reviewer 1 Report

This article presents an interesting and highly relevant topic to this special issue. The key findings and discussions are clearly established. Still, there are some concerns as follows:

The abstract may need to be rewrite. There is a potential logic problem as the conclusion “the nursing profession is building the competencies, knowledge and skills” is stated already in the background. Besides, the methodology might not be sufficiently reflected in the abstract.

The writing in the introduction is not connected well.

For the research question “how effective are the interventions”, it might not be sufficiently addressed in this work. Besides, it is not very sure if the word “intervention” covers all five prevention identified.

In section 3, the use of SIGN methodology can be further explained. Before discussion the various prevention identified, general comments developed from SIGN methodology can be added.

Other comments for considerations:

Shall the insights of nursing education be included in discussion so that it would even better fit in the theme of this special issue?

Author Response

First, we would like to thank you for your comments on the submitted manuscript.

Below, we respond to the comments made by the reviewers:

Comments and Suggestions for Authors

Response

Abstract

The abstract may need to be rewrite. There is a potential logic problem as the conclusion “the nursing profession is building the competencies, knowledge and skills” is stated already in the background. Besides, the methodology might not be sufficiently reflected in the abstract.

The summary is restructured to respond to the comment. Changes are shaded in yellow in the summary section.

Writing

The writing in the introduction is not connected well.

The connection between paragraphs in the introduction is reviewed. Likewise, the text has been revised by a native translator to review and improve the expression in the language of publication.

Research question

For the research question “how effective are the interventions”, it might not be sufficiently addressed in this work. Besides, it is not very sure if the word “intervention” covers all five-prevention identified.

The review protocol was based on the research question posed in the text. However, the research process itself, the studies found and the synthesis made, led us to conclusions that are within the topic but partially deviate from the original question. Therefore, following the instructions, we reformulated the research question. The question appears in the last paragraph of the Introduction section.

In section 3, the use of SIGN methodology can be further explained. Before discussion the various prevention identified, general comments developed from SIGN methodology can be added.

The interpretation of the levels of evidence established by SING is included in the "Summary of Results" section. This evaluation is included in the table to show the level of evidence for the interventions included in the review. None of the interventions has a negative classification (high risk of bias) and all of them could be used to establish recommendations.

Other comments for considerations:

Shall the insights of nursing education be included in discussion so that it would even better fit in the theme of this special issue?

The current special issue includes items such as "nursing practice" and "Competency development and evidence". From the authors' perspective, the review evidences the existence of nursing competencies in the management and prevention of bullying.

Reviewer 2 Report

Referees report on Nursing Interventions Against Bullying:

The mss is a summary of 17 articles on the role school nurses can and do play in reducing bullying among students.  Roughly half the articles were quantitative, half qualitative.  Papers in Spanish, English, and Portuguese were considered. The authors state they did no formal meta-analysis because of the disparity of methods employed by the articles.  The mss contains no original research.

The paper concludes that school nurses carry out interventions aimed at bullies, victims, and bystanders.  The research is competent, but the results are underwhelming.  We learn nothing about which interventions are most effective, and are in no position to do so, given their methodology.  What we learn is that more research is needed.  True enough, but one is left unsatisfied with this as the primary conclusion. 

Consider the following statement from the conclusion.  “the nurse may be able to judge when there is a need for referral to other professionals for more specialised treatment and even work holistically with other professionals to provide integrated care for all those affected by bullying.”  True enough, but we learn almost nothing about how frequently this occurs, or how it might be fostered. 

Author Response

Dear reviewer, first of all thank you for the review work done. We agree with the comments made. We hope that the changes made respond to the revisions made.

Response:

Comments and Suggestions for Authors

Response

Conclusion

The paper concludes that school nurses carry out interventions aimed at bullies, victims, and bystanders.  The research is competent, but the results are underwhelming.  We learn nothing about which interventions are most effective, and are in no position to do so, given their methodology.  What we learn is that more research is needed.  True enough, but one is left unsatisfied with this as the primary conclusion.

Consider the following statement from the conclusion.  “the nurse may be able to judge when there is a need for referral to other professionals for more specialised treatment and even work holistically with other professionals to provide integrated care for all those affected by bullying.”  True enough, but we learn almost nothing about how frequently this occurs, or how it might be fostered.

The last paragraph of the conclusion is clarified to address the highlights.

Reviewer 3 Report

The manuscript consists of total 12 pages, including 1 figure, 2 tables and the list of total 30 literature references. The article presents as a review of the problem of the nurses confronted with the problem of bullying at schools. Social pathology medical manifestations are current and important problem so the article is interesting for the Readers and consistent with the scope of works published by the Journal. The title is consistent with the contents of the manuscript, clear and informative enough. The English language used is acceptable.

The Abstract is structured but way too much focused on the methodology (more than half of the contents) instead providing the Readers with more information about the acutal axial results of the study, which would encourage them to read the full text.

The Introduction provides enough background information on the researched problem and justification for the study.

The Material and methods section is detailed enough.

The Results are clearly presented in a structured table form.

The Discussion is logically structured according to the type of prevention, roles and actors.

The Conclusions are comparably long but it is justified with the big scope of the results. The last 2 sentences that contain the actual call for action based on the study results are quite difficult to understand and differ strikingly in style from the previous clear and informative text, thus they shall be restated accordingly. The Authors may consider adding some discussion what actually shall be expected from a nurse and a remark that not all activities of nurses that were listed in the literature must in fact be fulfilled by the nurses (in the sense of their scope of professional activities); it is worth it to attempt to draw the line between the activities resulting directly from the professional role of the nurse (which is expected to be fulfilled by all nurses) and the role of the nurse as a conveniently due to profession located citizen engaged in positive pro-society activism (which is encouraged but not expected to be obligatory for all nurses). In many countries there is the tendency to shift all kinds of pro-society activities, most often demanded pro-bono, into the duty field of medical professionals, which may feel high spirited but in fact is neither logical nor doable for the already overloaded with work and responsibilities medical professionals.

The literature references are numerous and recent enough.

The Authors may consider including in their article the following aspects:
- comparison with other similar works in the field, e.c. https://doi.org/10.3390/ijerph20021577 https://doi.org/10.3390/healthcare10101835

Author Response

Dear reviewer, first of all we would like to thank you for the arduous review process. The contributions you request are interesting and coherent. Below we try to respond to the comments made.

Response

Comments and Suggestions for Authors

Response

Abstract

The Abstract is structured but way too much focused on the methodology (more than half of the contents) instead providing the Readers with more information about the acutal axial results of the study, which would encourage them to read the full text.

The results section of the summary is rewritten as indicated.

The Conclusions are comparably long but it is justified with the big scope of the results. The last 2 sentences that contain the actual call for action based on the study results are quite difficult to understand and differ strikingly in style from the previous clear and informative text, thus they shall be restated accordingly. The Authors may consider adding some discussion what actually shall be expected from a nurse and a remark that not all activities of nurses that were listed in the literature must in fact be fulfilled by the nurses (in the sense of their scope of professional activities); it is worth it to attempt to draw the line between the activities resulting directly from the professional role of the nurse (which is expected to be fulfilled by all nurses) and the role of the nurse as a conveniently due to profession located citizen engaged in positive pro-society activism (which is encouraged but not expected to be obligatory for all nurses). In many countries there is the tendency to shift all kinds of pro-society activities, most often demanded pro-bono, into the duty field of medical professionals, which may feel high spirited but in fact is neither logical nor doable for the already overloaded with work and responsibilities medical professionals.

The literature references are numerous and recent enough.

The Authors may consider including in their article the following aspects:

- comparison with other similar works in the field, e.c. https://doi.org/10.3390/ijerph20021577 https://doi.org/10.3390/healthcare10101835

The final part of the conclusions is reformulated in an attempt to respond to the comments made.